# Optimization of Emergency Supplies Scheduling for Hazardous Chemicals Storage Considering Risk

**Jianfeng Lu** [1] , **Xiaoxia Wang** [2,\*] **and Jiahong Zhao** [2]

1   Business Administration College, Nanchang Institute of Technology, Nanchang 330108, China;
    jflu2000@126.com
2   College of Civil and Traffic Engineering, Guangdong University of Technology, Guangzhou 510006, China;
    zhaojiahong1@126.com
\*   Correspondence: wxx@gdut.edu.cn

**Abstract:** Hazardous chemicals are harmful to the people around them during their storage, especially when an accident occurs. The allocation and scheduling of emergency materials, therefore, is an important imperative of emergency rescue services. Due to the harmful characteristics of dangerous goods, the storage risk of hazardous chemicals in emergency networks always exists, which threatens surrounding residents. To reduce the risk of hazardous chemicals storage in terms of emergency networks, the collaborative optimization of emergency materials allocation and scheduling from the perspective of risk is proposed in the present study. The risk assessment of dangerous goods storage in different stages is developed. Minimizing the total cost and risk, a bi-level programming model of emergency material allocation and scheduling for dangerous goods storage is formulated. Then, the Karush–Kuhn–Tucker (KKT) condition is introduced to transform the proposed model, and the solution method is designed based on an augmented $\varepsilon$-constraint method. Finally, the computational case is provided to demonstrate the workability of the proposed model and method.

**Keywords:** hazardous chemicals storage; emergency supplies scheduling; risk; bi-level planning; multi-objective optimization; augmented $\varepsilon$-constraint method

## 1. Introduction

Dangerous goods are harmful to their surroundings when there is an accident during storage. In recent decades, the storage of dangerous goods has resulted in frequent accidents, which has brought serious harm to the natural environment and human life. For example, on 12 August 2015, a huge fire and explosion occurred in a dangerous goods warehouse in Tianjin Port, located in Binhai New Area, Tianjin, which killed 165 people and caused direct economic losses of billions. Thus, emergency material scheduling is very important to reduce losses resulting from accidents.

The proper dispatching of emergency materials is very important in order to reduce accident-related losses. The problems facing emergency resource dispatching include how to select emergency service points and the quantity of materials to be deployed, planning of emergency routes, the selection of transportation modes, vehicle allocation arrangements as well as other issues. According to the model objectives, it mainly includes the minimum transportation cost as the objective, the minimum emergency response time, the minimum disaster losses, etc. [1–3]. Rathi [4] studied the problem of transporting large quantities of relief supplies or personnel between different supply and demand points of relief supplies in the event of a disaster, and the model was aimed at minimizing the quantity of unsatisfied goods. Ozday K. [5] studied a multi-commodity dynamic emergency resource allocation and scheduling problem, integrated the material network flow and vehicle routing problem, and established a mixed integer programming model with the objective of minimizing the number of unsatisfied emergency resources in each period. Zhang et al. [6] put forward the problem of multi-material allocation at accident



points, and designed an improved genetic algorithm using a mutation mechanism, a binary space partition tree, to solve this problem. Mohammadi R. [7] proposed a multi-objective stochastic programming model to create the earthquake emergency material scheduling plan, while Xing H. et al. [8] established a multi-objective planning model for earthquake emergency materials dispatching. Tzeng [9] designed an emergency resource transportation system, established a material distribution model by using a fuzzy multi-objective planning method, and analyzed the actual relief effect. Yuan et al. [10] set up a multi-objective optimization model for disaster relief material dispatch with the objective of minimizing cost and time, and solved it by the ideal point method. Sun Y. et al. [11] proposed a dual-objective emergency logistics scheduling model with two objectives, transportation time and transportation cost. The uncertainty of the model is reflected in two aspects: the occurrence time of emergencies, and the traffic volume predicted by the cloud model. X Li et al. [12] studied the location-routing problem of emergency logistics centers and material demand points, and established a multi-objective integer programming model based on the actual situation. The minimum total transportation time and the maximum total emergency material satisfaction were two objectives of the model. Some scholars have also studied dynamic emergency resource scheduling: Ren et al. [13] established a multi-material and multi-period dynamic scheduling model in a specific transportation network, and solved the model by hybrid genetic algorithm, while Wang W. et al. [14] established an optimization model of emergency material dispatch with the objective of the highest reliability.

Due to the special characteristics of emergency materials dispatching for dangerous goods storage accidents, the above research results are not fully applicable. Dangerous goods are flammable, explosive, poisonous, and harmful. In the process of emergency rescue, because of the untimely supply of emergency materials, the storage of hazardous chemicals in the emergency network threatens the surrounding residents and poses risks.

In the face of state-of-the-art works related to the risk assessment of hazardous materials storage and transportation, we make a comparison and report the results in Table 1. As can be seen, four items are involved, where the column named "hazmat" is the type of hazardous materials considered, the "model" column reports the different methods applied to assess risks, and "type" indicates the type of proposed risk assessments. It can be concluded that, in comparison to the qualitative assessment, the quantitative method can estimate the consequences following accidents.

**Table 1.** Comparisons of the risk assessment.

| Author (Year) [Ref] | Hazardous Materials | Model | Type |
|---|---|---|---|
| Crawley et.al (2003) [15] | Oil and gas | Failure modeling | Quantitative |
| Young-Do & Bum (2005) [16] | Natural gas | Fatal length | Quantitative |
| Young-Do & Bum (2006) [17] | Hydrogen gas | Affected area | Quantitative |
| Spyros & Fotis (2006) [18] | Fuel gas | Safety distance | Quantitative |
| Shebeko et al. (2007) [19] | Oil | Failure rate | Qualitative |
| Tim et al. (2012) [20] | Oil and gas | Evaluate current risk assessments | Qualitative |
| Thomas et al. (2012) [21] | Crude oil | Risk analysis | Qualitative |
| Petrissa & Peter (2013) [22] | Oil | Bayesian data analysis | Qualitative |
| Li et al. (2017) [23] | NG | Pipeline instability | Quantitative |

Therefore, it is necessary to consider the risk of missing emergency supplies and optimize the emergency supplies dispatching plan for hazardous chemicals storage accidents. In actual production application, the emergency logistics system is complex, and involves integration of the guidance and decision-making of both the upper government departments as well as the operational decision-making of the lower emergency command department. According to the bi-level programming principle, this paper combines and optimizes the facility location, capacity allocation and cross-level resource scheduling of hazardous chemicals storage emergency logistics system. Optimization is divided into two

stages: pre-emergency planning and post-emergency operation. The pre-stage mainly involves the decision-making of facility location and capacity allocation, while the post-stage mainly involves the allocation and scheduling of emergency materials.

## 2. Problem Description

As shown in Figure 1, the government emergency management department is set as the upper decision-making level, and the emergency command center is set as the lower decision-making level.

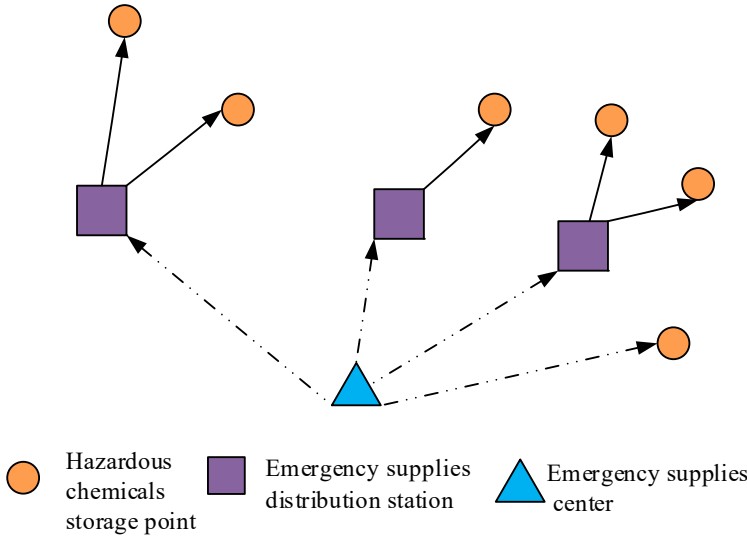

**Figure 1.** Schematic diagram of multi-stage emergency logistics system.

By optimizing the construction position of emergency material distribution station, the upper layer affects the supply and dispatch of emergency materials in the lower layer, while optimizing the distribution and supply routes of emergency materials allows the feedback from the lower management layer to influence the adjustment of the upper emergency material distribution station in the later period.

The risk of the hazardous chemicals storage can be formulated as follows:

$$R_i = \sum_l (p_i \times m_i)^{K_l} \tag{1}$$

where $R_i$ is A is the risk of missing emergency supplies in hazardous goods storage at accident point $i$, $p_i$ is the number of residents in the vicinity of the accident point $i$, $m_i$ is the demand for emergency supplies at the accident point $i$, and $K_l$ is the risk perception coefficient in stage $l$.

When the risk perception coefficient is less than 1, the residents' perception of risk is low, and they are optimistic. When the risk perception coefficient is equal to 1, the residents' perception of risk is normal and belongs to the middle attitude. When the risk perception coefficient is greater than 1, residents have a high degree of perception of risks, which is a pessimistic attitude.

## 3. Problem Formulation

### 3.1. Assumptions

The model is based on the following assumptions:

(1)  In the process of distribution of emergency materials, the influence of other random factors, such as road flows restriction, is not considered;
(2)  There are linear paths between nodes in the network;
(3)  All sections carry at the same speed;

(4)  The loading capacity of transport vehicles meets the requirements;
(5)  There is only one type of hazardous chemicals storage.

*3.2. Notations*

Sets, parameters, and decision variables of the model are described as follows:

3.2.1. Set

$S$ set of emergency material demand point, $S = (1, 2, \cdots, s)$;
$F$ set of emergency materials distribution station, $F = (1, 2, \cdots, f)$;
$\overline{F}$ set of emergency materials center, $\overline{F} = (1, 2, \cdots, \overline{f})$;
$L$ set of type of levels for each of the capabilities, $L = (1, 2, \cdots, l)$.

3.2.2. Parameters

$C_{il}^{fix}$ fixed construction costs of distribution station $i$ on level $l$;
$C_{il}^{var}$ the unit variable cost of distribution station $i$ on level $l$;
$C_{ijl}^{all-p}$ the cost of the delivery service provided by distribution station $i$ on level $l$ to the demand point $j$;
$C_{ijl}^{all-b}$ the cost of emergency material center $i$ to provide resupply services to distribution station $j$ on level $l$;
$C_{ijl}^{tra-p}$ the unit transportation cost of distribution service provided by distribution station $i$ for demand point $j$ on level $l$;
$C_{ijl}^{tra-b}$ the unit transport costs of emergency material center $i$ to provide resupply services to distribution station $j$ on level $l$;
$q_i$ the demand for emergency materials at demand point $i$;
$m_i$ the total amount of hazardous chemicals stored at demand point $i$;
$p_i$ the number of residents at demand point $i$;
$D_{ij}$ the distance of section $i, j$;
$T_{ij}$ the transportation time of section $i, j$;
$A_{il}$ the maximum material storage capacity of Level $l$ distribution station $i$;
$v$ the average running speed of transport vehicles;
$H$ the expected transportation time of emergency materials;
$K_i$ the risk perception coefficient of residents at the demand point $i$;
$M$ An infinite positive integer.

3.2.3. Decision Variables

$o_{il}$ *1* If emergency supplies distribution station $i$ with configuration level $l$ is built; *0* otherwise;
$x_{ijl}$ *1* if emergency supplies distribution station $i$ with configuration level $l$ provide services to demand point $j$; *0* otherwise;
$y_{ijl}$ *1* if emergency supplies center $i$ provides services to emergency supplies distribution station $j$ with configuration level $l$; *0* otherwise;
$x\prime_{ijl}$ The number of emergency resources delivered by emergency supplies distribution station $i$ with configuration level $l$ to demand point $j$;
$y\prime_{ijl}$ The number of emergency supplies provided by emergency supplies center $i$ to emergency supplies distribution station $j$ with configuration level $l$;
$z_{il}$ The number of emergency supplies available for distribution at emergency material distribution station $i$ with allocation level $l$.

*3.3. Formulation*

(1)  upper-level planning model

$$\min h_1 = \sum_{i \in F} \sum_{l \in L} C_{il}^{fix} o_{il} + \sum_{i \in S} \sum_{j \in F} \sum_{l \in L} C_{ijl}^{all-p} x_{ijl} + \sum_{i \in \overline{F}} \sum_{j \in F} \sum_{l \in L} C_{ijl}^{all-b} y_{ijl} \qquad (2)$$

$$\min h_2 = \sum_{i \in S} \sum_{j \in F} \sum_{l \in L} \left( p_i \times m_i \times x_{ijl} \right)^{K_i} \tag{3}$$

s.t.,

$$K_i = \begin{cases} 0.5, & T_{ij} < 0.5 \times H \\ 1, & 0.5 \times H \le T_{ij} \le H \\ 1.5, & H < T_{ij} \le 2H \\ 2, & T_{ij} > 2H \end{cases} \qquad \forall i \in S, \forall j \in F \tag{4}$$

$$T_{ij} = \frac{D_{ij}}{v} \qquad \forall i \in S, \forall j \in F \tag{5}$$

$$\sum_{j \in S} x_{ijl} \le M \times o_{il} \qquad \forall i \in F, \forall l \in L \tag{6}$$

$$\sum_{j \in S} x_{ijl} \times q_j \le A_{il} \times o_{il} \qquad \forall i \in F, \forall l \in L \tag{7}$$

$$\sum_{i \in F} \sum_{l \in L} x_{ijl} \ge 1 \qquad \forall j \in S \tag{8}$$

$$\sum_{i \in \overline{F}} y_{ijl} \le M \times o_{jl} \qquad \forall j \in F, \forall l \in L \tag{9}$$

$$\sum_{i \in \overline{F}} \sum_{l \in L} y_{ijl} \ge 1 \qquad \forall j \in S \tag{10}$$

$$\sum_{l \in L} o_{il} = 1 \qquad \forall i \in F \tag{11}$$

$$\sum_{i \in F} \sum_{l \in L} o_{il} \ge 1 \tag{12}$$

$$o_{il} = 0, 1, \ \forall i \in F, \forall l \in L \tag{13}$$

$$x_{ijl} = 0, 1, \ \forall i \in F, \forall l \in L, \ \forall j \in S \tag{14}$$

$$y_{ijl} = 0, 1, \ \forall i \in \overline{F}, \forall l \in L, \ \forall j \in F \tag{15}$$

(2) Lower-level planning model

$$\min h_3 = \sum_{i \in F} \sum_{l \in L} C_{il}^{\mathrm{var}} z_{il} + \sum_{i \in S} \sum_{j \in F} \sum_{l \in L} C_{ijl}^{tra-p} D_{ij} x\prime_{ijl} + \sum_{i \in \overline{F}} \sum_{j \in F} \sum_{l \in L} C_{ijl}^{tra-b} D_{ij} y\prime_{ijl} \tag{16}$$

$$\min h_4 = \sum_{i \in S} \sum_{j \in F} \sum_{l \in L} \left( p_i \times m_i \times \frac{x\prime_{ijl}}{q_i} \right)^{K_i} \tag{17}$$

s.t.,

$$K_i = \begin{cases} 0.5, & T_{ij} < 0.5 \times H \\ 1, & 0.5 \times H \le T_{ij} \le H \\ 1.5, & H < T_{ij} \le 2H \\ 2, & T_{ij} > 2H \end{cases} \qquad \forall i \in S, \forall j \in F \tag{18}$$

$$T_{ij} = \frac{D_{ij}}{v} \qquad \forall i \in S, \forall j \in F \tag{19}$$

$$\sum_{i \in F} \sum_{l \in L} x\prime_{ijl} \ge q_j \qquad \forall j \in S \tag{20}$$

$$x\prime_{ijl} \le M \times x_{ijl} \qquad \forall i \in F, \forall j \in S, \forall l \in L \tag{21}$$

$$\sum_{j \in S} x\prime_{ijl} \le A_{il} \times x_{ijl} \qquad \forall i \in F, \forall l \in L \tag{22}$$

$$\sum_{j \in S} y\prime_{ijl} \le z_{il} \qquad \forall i \in F, \forall l \in L \tag{23}$$

$$z_{jl} = \sum_{i \in \overline{F}} y\prime_{ijl} \qquad \forall j \in F, \forall l \in L \qquad (24)$$

$$y\prime_{ijl} \leq M \times x\prime_{ijl} \qquad \forall i \in \overline{F}, \forall j \in F, \forall l \in L \qquad (25)$$

$$\sum_{i \in \overline{F}} y\prime_{ijl} \leq A_{jl} \times x\prime_{ijl} \qquad \forall j \in F, \forall l \in L \qquad (26)$$

$$z_{il} \geq 0, \forall i \in F, \forall l \in L \qquad (27)$$

$$x\prime_{ijl} \geq 0, \forall i \in F, \forall l \in L, \forall j \in S \qquad (28)$$

$$y\prime_{ijl} \geq 0, \forall i \in \overline{F}, \forall l \in L, \forall j \in F \qquad (29)$$

### 3.4. Equations Explanation

The upper-level planning model mainly optimizes the content of facility location and capacity allocation. Equations (2) and (3) are objective functions, which represent the minimization of total cost and the minimization of total risk, respectively.

The total cost expressed by Equation (2) includes: fixed construction cost and capacity allocation cost of emergency material distribution station, distribution cost of emergency material distribution services between emergency material distribution stations, as well as the demand point and distribution cost of emergency material replenishment services between the emergency material center and distribution station.

The risk expressed by Equation (3) includes the different perceived risks of residents at each demand point when facing the threat from hazardous chemicals stock.

Constraint (4) indicates the perceived risk assessment, which shows that perceived risk coefficient is closely related to emergency time. Constraint (5) is the calculation formula of road transportation time, which is expressed as the ratio of road length to average running speed of vehicles. Constraint (6) is the logical constraint of decision variables, which means that only the emergency material distribution stations which have been built and assigned a capacity can provide emergency material distribution services for the demand points. Constraint (7) is the capacity constraint of emergency material distribution stations, which means that the total demand of any emergency material distribution station at the demand point of the distribution service cannot exceed the capacity allocation level of the emergency material distribution station. Constraint (8) indicates that any demand point will be provided with a resource distribution service by at least one emergency material distribution station.

Constraint (9) is the logical constraint of decision variables, which indicates that the emergency material center can only provide an emergency material supply service for emergency material distribution stations which are determined to be built and have the capacity distribution level.

Constraint (10) indicates that any emergency material distribution station will be supplied by at least one emergency material center. Constraint (11) indicates that any emergency material distribution station can only be configured with one capacity level. Constraint (12) indicates that at least one emergency material distribution station is built in the network. Constraints (13)–(15) are the definition fields of decision variables.

The lower-level planning model mainly optimizes the quantity distribution of emergency supplies and other supplies. Equations (16) and (17) are objective functions, which represent the minimization of total cost and total risk, respectively.

The total cost expressed by Equation (16) includes: the variable cost of emergency material distribution station, emergency material distribution, and transportation cost between the emergency material distribution station and demand point, as well as the emergency material replenishment and transportation cost between emergency material center and distribution station. The risk expressed by Equation (17) includes the different perceived risks of residents at each demand point when facing the existing threat of hazardous chemicals stock.

Constraint (18) is the calculation formula of perceived risk coefficient, which shows that perceived risk coefficient is closely related to emergency time, and the longer the actual emergency time, the higher the perceived risk coefficient. Constraint (19) is the calculation formula of road section transportation time, which is expressed as the ratio of road section length to average vehicle running speed. Constraint (20) shows that the total amount of emergency materials distributed by the emergency material distribution station to each demand point should meet its demand. Constraint (21) is the logical constraint of decision variables, which means that only emergency material distribution stations that have determined the distribution relationship of resource distribution services can provide services for demand points. Constraint (22) is the capacity constraint of emergency material distribution station, which means that the total demand of any emergency material distribution station at the demand point of distribution service cannot exceed the capacity allocation level of the emergency material distribution station. Constraint (23) indicates that the resources that can be distributed by any emergency material distribution station cannot exceed the amount of supply it has obtained. Constraint (24) is a method for calculating the resource replenishment of any emergency material distribution station, i.e., the total resources owned by the emergency material distribution station come from the replenishment of the emergency material center. Constraint (25) is the logical constraint of decision variables, which indicates that the emergency material center can only provide an emergency material supply service for emergency material distribution stations which are determined to be built and have the capacity distribution level. Constraint (26) indicates that the supply quantity received by any emergency material distribution station cannot exceed its maximum carrying capacity. Constraints (27–29) are the definition fields of decision variables.

## 4. Solution Procedure

### 4.1. Description of Solution Method

For the bi-level programming model, the model can be transformed by the KKT (Karush–Kuhn–Tucker) condition, and the nonlinear objective function can be linearized. Thus, the multi-objective model can be transformed into a single-objective model by multi-objective optimization method, and then solved by optimization software. Technical methods involved in solving specific implementation steps are as follows:

(1)  KKT conditional transformation.

In solving the bi-level programming model, we can consider transforming it into a single-layer model in an appropriate way, and then use the conventional solution method to solve the model. In this paper, the KKT conditional method will be used to complete the transformation of the bi-level model. The KKT condition is one of the common methods for solving optimization problems; in the proposed bi-level programming model, the KKT condition can transform the lower model into this kind of condition to solve it. The general bi-level programming model has the following forms:

$$\min F(x) \tag{30}$$

$$\min f(x) \tag{31}$$

s.t.,

$$h_j(x) = 0, j = 1, 2, \ldots p \tag{32}$$

$$g_k(x) \leq 0, k = 1, 2 \ldots p \tag{33}$$

which can be transformed as:

$$\min F(x) \tag{34}$$

s.t.,

$$\left. \frac{\partial L}{\partial X} \right|_{x=x_0} = 0 \tag{35}$$

$$\lambda_j \neq 0 \tag{36}$$

$$\mu_k \geq 0 \tag{37}$$

$$\mu_k g_k(x_0) = 0 \tag{38}$$

$$h_j(x_0) = 0 \tag{39}$$

$$g_k(x_0) = 0 \tag{40}$$

The bi-level programming model can be transformed into a single-level programming model by the KKT conditional method.

(2)  Linear transformation.

After only using the KKT condition to transform the model, the model still has multiple objective functions, and the risk objective function is nonlinear. Before using the KKT condition, the model should be linearized. In the upper-level planning model constructed in this chapter, the associated decision variables of the risk function are 0–1 variables, thus there is no need to consider the linear transformation problem, and the risk objective function in the lower-level model is mainly treated linearly.

The value of the risk perception coefficient is {0.5, 1, 1.5, 2}.

$$Risk = \sum_{i \in S}(p_i \times m_i)^{K_i} \tag{41}$$

Condition 1: when $_i$ = 1, Equation (41) is

$$Risk = \sum_{i \in S} p_i \times m_i \tag{42}$$

which is a linear formula, without conversion.

Condition 2: when $K_i$ = 0.5, Equation (41) is

$$Risk = \sum_{i \in S}(p_i \times m_i)^{\frac{1}{2}} = \sum_{i \in S} \sqrt{(p_i \times m_i)^2} \leq \sum_{i \in S}(p_i \times m_i) \tag{43}$$

it can be linearized by inequality transformation.

Condition 3: when $Ki$ = 1.5, formula (41) is

$$Risk = \sum_{i \in S}(p_i \times m_i)^{\frac{3}{2}} = \sum_{i \in S} \sqrt[3]{(p_i \times m_i)^2} \tag{44}$$

Equation (44) can be approximated by linear transformation through unequal transformation,

$$Risk = \sum_{i \in S} \sqrt[3]{(p_i \times m_i)^2} \leq \sum_{i \in S}(p_i \times m_i) \tag{45}$$

Condition 4: when $Ki$ = 2, Equation (41) is

$$Risk = \sum_{i \in S}(p_i \times m_i)^2 \tag{46}$$

(3) Multi-objective optimization.

Through the above steps, the transformation of the bi-level programming model and the linear transformation of the nonlinear objective function are processed in turn. At present, the model has four objective functions with different metrics, in which the cost objective function and the risk objective function both have the same measurement unit, which can be directly integrated by setting the weight coefficient. Therefore, there are two optimization objectives with different measurement units. The method of augmented $\varepsilon$-constraint [24–26] can be adopted, which focuses on cost objectives, generating different

upper and lower bounds by constantly cutting down the optimization interval of risk objectives as well as optimizing step-by-step to obtain the optimal solution.

According to the augmented-constraint method proposed by Mavrotas et al., a solution method based on the augmented $\varepsilon$-constraint method is designed. In the augmented-constraint method, a relaxation variable $\mu_2$ and a sufficiently small normal number *eps* are introduced, and an extra term is added to the minimum cost objective, that is, the relaxation variable $\mu_2$ is multiplied by the range $r_2$ of the risk objective function, and the objective is constrained by the relaxation variable and constraint coefficient. The dictionary order method is used to calculate the target value of the model, and the form of the transformed single-objective model is as follows:

$$
\begin{aligned}
\min \quad & (h_1 - eps \times \mu_2/r_2) \\
s.t \quad & h_2 + \mu_2 = \varepsilon \\
& \mu_2 \geq 0 \\
& and\ existing\ domain
\end{aligned}
\tag{47}
$$

In this formula, $h_2$ is the cost objective function, $h_2$ is the risk objective function, $r_2$ is the range of the risk objective function, $\mu_2$ is the minimum normal number, and $\varepsilon$ is the constraint level of the total risk objective.

It is also worth noting that the model can be solved by conventional optimization software.

### 4.2. Solving Steps

The model is transformed according to KKT condition, linearized and solved by multi-objective optimization method as follows:

Step 1: Define sets and parameters and input relevant data;

Step 2: according to KKT conditions, the bi-level programming model is transformed into a single-level programming model;

Step 3, setting weight coefficients $\lambda_1$ and $\lambda_2$ of two cost objective functions, and carrying out linear weighted integration on the cost objective functions;

Step 4, setting weight coefficients $\lambda'_1$ and $\lambda'_2$ of two risk objective functions, and carrying out linear weighted integration on the risk objective functions;

Step 5: Introducing parameter $a_i$ to linearize and approximate linearize the integrated risk objective function;

Step 6: Use Cplex to solve the objective functions when minimizing and maximizing the risk sub-objectives, and get the optimal values of $h'_2{}^{min}$ and $h'_2{}^{max}$;

Step 7: Introduce relaxation variable *eps* and coefficient $\mu$, and add wide-area increasing constraint to the multi-objective single-level programming model, so that it changes into the single-objective single-level linear programming model in the form of Equation (46);

Step 8: Using Cplex to solve that single-objective single-layer linear programming model in the form of Equation (47), and obtain the optimal solution.

## 5. Numerical Example

### 5.1. Basic Information

As shown in Figure 2, a test example with a total number of 30 network nodes is randomly generated within the range of $30 \times 30\ km^2$ with 16 hazardous chemical storage points (No.1–16). There are 10 candidate points for an emergency material distribution station (numbered 17–26). There are four existing resource centers (numbered 27–30). Each network node is connected with one another, and the distance between each node uses the linear distance of coordinates. The basic information of emergency material demand points is shown in Table 2, and the basic information of candidate points of each emergency material distribution station is shown in Table 3, while the replenishment cycle is assumed to be one year. The distribution cost of each type of emergency materials is CNY 200/t, and the supply service cost is CNY 150/t. The distribution unit transportation cost of each type of emergency materials is CNY 150/t·km, and the supply unit transportation cost is CNY 100/t·km. The average vehicle speed is 50 km/h. The value of the risk perception

coefficient of the upper model is 0.5, and that of the lower model is 1. Assuming that the region was attacked by terrorism, all hazardous chemicals storage sites had accidents, and it is necessary to distribute emergency materials to all storage sites.

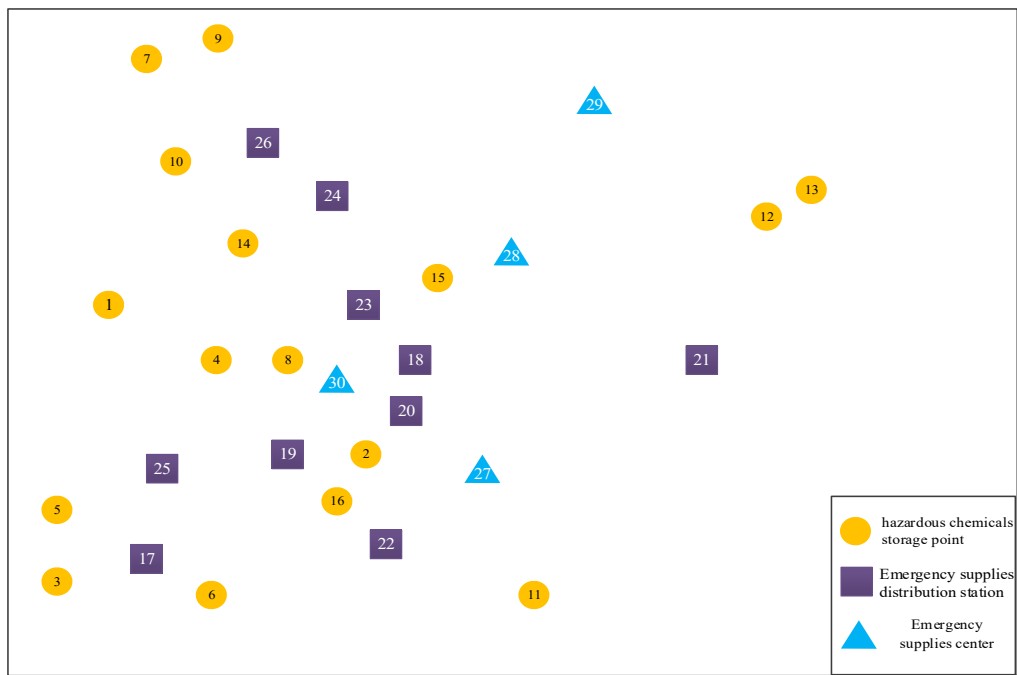

**Figure 2.** Schematic diagram of network nodes.

**Table 2.** Information of the hazardous chemicals storage points.

| Node | Population (Thousands) | Storage of Hazardous Chemicals (t) | Demand for Emergency Materials (t) |
|------|------------------------|------------------------------------|------------------------------------|
| 1 | 6.10 | 12 | 1.47 |
| 2 | 9.81 | 7 | 1.59 |
| 3 | 2.05 | 8 | 1.11 |
| 4 | 1.10 | 16 | 0.81 |
| 5 | 6.45 | 18 | 1.27 |
| 6 | 9.15 | 15 | 1.64 |
| 7 | 4.38 | 19 | 0.59 |
| 8 | 2.26 | 15 | 1.63 |
| 9 | 3.16 | 12 | 1.78 |
| 10 | 9.70 | 10 | 1.97 |
| 11 | 6.10 | 10 | 1.69 |
| 12 | 1.26 | 10 | 1.60 |
| 13 | 2.66 | 10 | 1.46 |
| 14 | 8.18 | 7 | 0.74 |
| 15 | 6.83 | 11 | 1.63 |
| 16 | 4.21 | 20 | 1.42 |

**Table 3.** Basic information of candidate points of emergency supplies distribution stations.

| Serial Number | Fixed Cost (10⁴ Yuan/Year) | Variable Cost (10⁴ Yuan/t.Year) | Facility Capacity (t/Year) (Level 1, Level 2, Level 3) |
|---|---|---|---|
| 17 | 140 | 35 | (5,10,15) |
| 18 | 101 | 28 | (15,20,25) |
| 19 | 125 | 44 | (5,10,15) |
| 20 | 102 | 31 | (15,20,25) |
| 21 | 132 | 34 | (5,10,15) |
| 22 | 144 | 43 | (15,20,25) |
| 23 | 30 | 33 | (15,20,25) |
| 24 | 37 | 35 | (20,25,30) |
| 25 | 38 | 34 | (20,25,30) |
| 26 | 30 | 33 | (20,25,30) |

*5.2. Calculation Results*

Under the computer Intel(P)/CPU2.2 GHz/2G environment, Java is used for programming, and Cplex version 12.10 is called for calculation. The value interval in the augmentation-constraint method is equally divided into 10 intervals, of which the value of the first interval is the single-objective value result with the smallest total cost. The results obtained in the first nine intervals are listed, as shown in Table 4 below.

**Table 4.** Calculation results of single interval by augmentation-constraint method.

| Interval Interval | Total Cost (10⁶ Yuan) | Total Risk (10³ Person·t) | Gap (%) |
|---|---|---|---|
| 1 | 442.67 | 879.80 | 11.24 |
| 2 | 476.03 | 634.28 | 10.47 |
| 3 | 500.29 | 364.74 | 3.79 |
| 4 | 540.98 | 204.88 | 1.70 |
| 5 | 572.35 | 177.29 | 6.04 |
| 6 | 599.48 | 167.29 | 16.46 |
| 7 | 630.56 | 146.71 | 14.45 |
| 8 | 663.89 | 140.53 | 17.36 |
| 9 | 684.31 | 189.22 | 16.99 |

It can be seen that within 3600 s, multiple effective solutions can be generated according to different value intervals, and the Gap values of the solution results are all less than 18%. Taking this calculation example, the result scheme with a value interval of 4 is recommended. The total cost of this scheme is CNY 540.98 × 10⁶, and the total risk is 204.88 × 10³ man tons. The corresponding facilities' locations, capacity allocation, vehicle distribution and replenishment can be seen in Table 5 and Figure 3.

**Table 5.** Recommended Scheme Table.

| Distribution Station (Location, Capacity) | Route | Quantity of Emergency Supplies (t) | Route | Emergency Materials Distribution Quantity (t) |
|---|---|---|---|---|
| (20, 20) | 27–20 | 3.01 | 20–2 | 1.59 |
| | | | 20–16 | 1.42 |
| (22, 20) | 27–22 | 3.33 | 22–6 | 1.64 |
| | | | 22–11 | 1.69 |
| (21, 15) | 28–21 | 3.06 | 21–12 | 1.6 |
| | | | 21–13 | 1.46 |

**Table 5.** *Cont.*

| Distribution Station (Location, Capacity) | Route | Quantity of Emergency Supplies (t) | Route | Emergency Materials Distribution Quantity (t) |
|---|---|---|---|---|
| (26, 25) | 29–26 | 4.34 | 26–10 | 1.97 |
| | | | 26–7 | 0.59 |
| | | | 26–9 | 1.78 |
| (23, 25) | 30–23 | 4.65 | 23–1 | 1.47 |
| | | | 23–4 | 0.81 |
| | | | 23–8 | 1.63 |
| | | | 23–14 | 0.74 |
| | | | 23–15 | 1.63 |
| (25, 20) | 30–25 | 4.01 | 25–3 | 1.11 |
| | | | 25–5 | 1.27 |

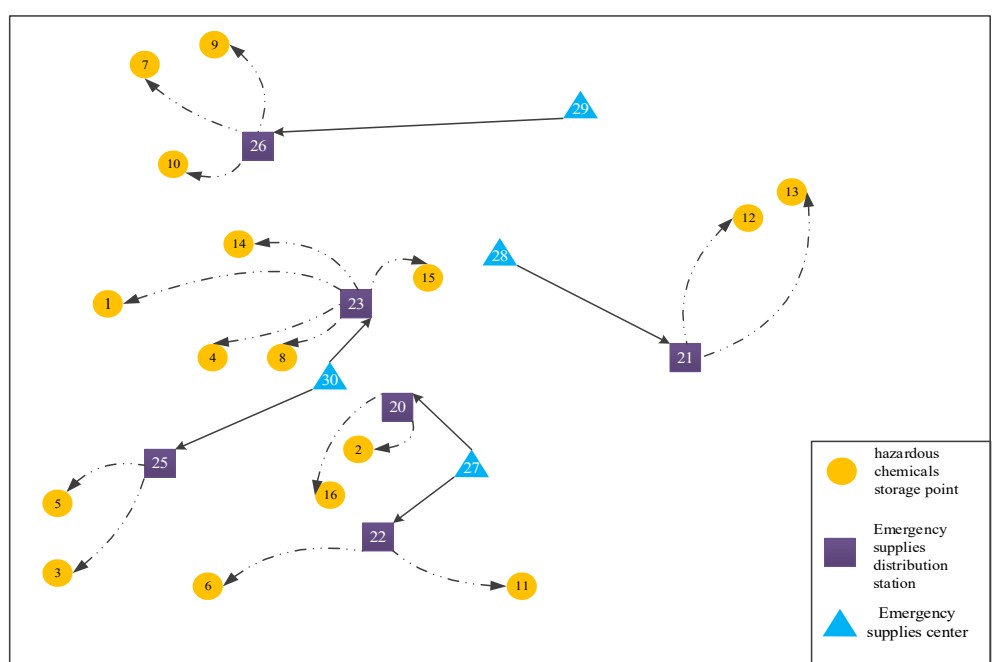

**Figure 3.** Schematic diagram of recommended scheme.

### 5.3. Stability Analysis

In order to verify the stability of the designed solution method, 50 groups of test cases are randomly generated on the basis of the basic information and calculation scale of Example 5.1, while the 16 coordinate positions of demand points are randomly set for each test case. In the established generation interval, the number of the surrounding population and the demand for emergency materials at each accident point are randomly generated. The calculation time is set to 3600 s, and the calculation efficiency of each test case is expressed by GAP value and calculation time, which are summarized in Table 6. According to the data in the table, the average GAP value of 50 groups of test cases is 16.59%, while the average calculation time of effective solution is 80.73 s.

**Table 6.** Computational Stability Analysis of Test Cases.

| Test | GAP (%) | Calculation Time (s) | Test | GAP (%) | Calculation Time (s) |
|------|---------|----------------------|------|---------|----------------------|
| 1 | 21.11 | 3595.6 | 26 | 16.93 | 985.32 |
| 2 | 19.61 | 2782.55 | 27 | 15.01 | 2781.54 |
| 3 | 11.85 | 3600.00 | 28 | 24.48 | 2646.90 |
| 4 | 13.68 | 3600.00 | 29 | 24.22 | 1395.36 |
| 5 | 1.77 | 3600.00 | 30 | 10.29 | 3577.14 |
| 6 | 13.99 | 903.95 | 31 | 14.54 | 2524.50 |
| 7 | 16.28 | 3600.00 | 32 | 13.11 | 3280.32 |
| 8 | 16.28 | 3600.00 | 33 | 18.39 | 1995.12 |
| 9 | 12.43 | 3600.00 | 34 | 14.54 | 1829.88 |
| 10 | 22.08 | 3600.00 | 35 | 22.30 | 777.24 |
| 11 | 13.68 | 3600.00 | 36 | 16.94 | 1787.04 |
| 12 | 10.60 | 3600.00 | 37 | 23.41 | 2812.14 |
| 13 | 21.80 | 1348.35 | 38 | 13.44 | 428.40 |
| 14 | 11.66 | 3600.00 | 39 | 18.17 | 1300.50 |
| 15 | 24.16 | 1550.35 | 40 | 12.08 | 2252.16 |
| 16 | 17.54 | 3600.00 | 41 | 23.28 | 3372.12 |
| 17 | 14.69 | 3600.00 | 42 | 12.90 | 3066.12 |
| 18 | 21.72 | 919.10 | 43 | 12.81 | 3173.22 |
| 19 | 26.91 | 3600.00 | 44 | 12.48 | 2368.44 |
| 20 | 1.10 | 3600.00 | 45 | 18.54 | 2249.10 |
| 21 | 17.22 | 2115.95 | 46 | 13.48 | 1377.33 |
| 22 | 12.08 | 3600.00 | 47 | 19.56 | 1802.34 |
| 23 | 22.36 | 3600.00 | 48 | 13.19 | 1906.38 |
| 24 | 17.25 | 3600.00 | 49 | 16.16 | 3366.91 |
| 25 | 23.89 | 2893.65 | 50 | 15.27 | 3427.2 |

## 6. Conclusions

The storage of hazardous chemicals has a high potential to cause personal injury and loss of property. In order to ensure the effectiveness of emergency rescue, the dispatch of emergency supplies is very important following the occurrence of many hazardous chemicals storage accidents. Due to the flammable, explosive, toxic, and harmful characteristics of hazardous chemicals, the storage risk of hazardous chemicals always exists in the process of emergency rescue. This paper presents a bi-level programming model introduced for emergency supplies scheduling for hazardous chemicals storage which considers risk. In this study, the optimization of emergency logistics resource system is divided into the early stage of facility location–capacity allocation planning and the late stage of supply–distribution route design.

The main conclusions are as follows:

(1) According to the different stages of emergency logistics management, the risk measurement model of dangerous goods storage in different emergency stages is put forward.

(2) Through incorporation of the perceived risk differences in different emergency stages into risk objectives, a bi-level programming model for emergency material allocation and scheduling of hazardous chemicals storage with minimum risk and minimum cost is constructed, and the KKT conditions are adopted to transform the model; further, a solution method based on augmented $\varepsilon$-constraint method is designed.

(3) An example verifies the effectiveness of the model and solution method designed in this paper. In the stability test, the solution method designed in this paper can obtain a stable and effective solution in a given time. Compared with the traditional optimization model with the shortest total emergency time as the index, the optimization model with risk as the goal can reduce the total cost and total risk of emergency logistics system by about 19.6% and 11.3%, respectively.

The paper contributes practical research results, while the research results provide a solution for the allocation and scheduling of emergency supplies in chemical accidents, and the proposed scheme can minimize the cost and risk associated with emergencies.

In this study, the proposed model has strong applicability, and the algorithm is simple to calculate, which can help decision makers quickly determine the emergency material scheduling plan after the accident.

However, our paper has limitations, and can be extended in several directions. Firstly, the data from the case study are not the real data of the actual emergency rescue work, and the solution is not convincing enough. Further research would collect more real data and determine solutions in less time by developing more efficient algorithms. Furthermore, the established optimization model only considers the two objectives of cost and risk. Further research would introduce more objectives so that the model can better describe the real situation. The third possible outcome of further research is the time window constraint incorporated into our problem, and develop practically implementable solution models and algorithms.

**Author Contributions:** All authors contributed equally to this work. In particular, J.L. put forward the initial idea of research, designed the research method and drafted the first draft. X.W. performed the case study. J.Z. revised the manuscript. All authors have read and agreed to the published version of the manuscript.

**Funding:** This research was funded by Chinese National Natural Science Fund grant number 61803091, 61803092 and 71801052.

**Institutional Review Board Statement:** No applicable.

**Informed Consent Statement:** No applicable.

**Data Availability Statement:** No applicable.

**Acknowledgments:** This research is supported by the Project of Chinese National Natural Science Fund (Serial no. 61803091, 61803092, 71801052).

**Conflicts of Interest:** The authors declare no conflict of interest.

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
