# Peer review of "Optimization of Emergency Supplies Scheduling for Hazardous Chemicals Storage Considering Risk"

_sustainability, doi:10.3390/su131910718_

Round 1
Reviewer 1 Report
Dear Authors:
This is an interesting study but I don't think that your manuscript is appropriate for the readership of this journal. This is a good read for OR outlets with a focus on mathematical modeling and computation and your selection of references supports this recommendation.
Best wishes.
Author Response
Dear reviewer,
Thank you very much for your carefully work, and we do appreciate your work.
Please find the attachment of the mail, which is the reply to the comments.
According to your comments and suggestions, we have revised the manuscripted.
Thanks a lot, again!
Best,
Jianfeng Lu and Jiahong Zhao

Reviewer 2 Report
In the attached file you will find the aspects to improve the work to continue with the process

Author Response

(The authors gave the same response as above.)

Reviewer 3 Report
The reviewer finds several important information from the manuscript which tried to establish and optimize emergency logistics resource system. The reviewer recognized that the system was optimized by using several analyses and the examples.
The reviewer recommends the authors to further elaborate what applicability the system is and how the optimized system can be applied for a actual case and to discuss effectiveness with the optimized system because many readers would want to know it.
Author Response

(The authors gave the same response as above.)

Round 2
Reviewer 1 Report
Thank you for your efforts. The paper now seems much improved in its current form.
Author Response
English spelling was checked
Reviewer 2 Report
In the paper, even when the Roman numerals have been changed to Arabic numerals, the citations continue to appear as a subscript and not in the stipulated font size, which is the same as in the text.
Author Response
The formatting of the citation has been fixed